# Modelling and Experiment of an Adjustable Device Combining an Inerter and a Damper

**Xiaoliang Zhang [1,\*], Weian Zhu [1] and Jiamei Nie [2]**

[1]   Automotive Engineering Research Institute, Jiangsu University, 301 Xuefu Road, Zhenjiang 212013, China
[2]   School of Automotive and Traffic Engineering, Jiangsu University, 301 Xuefu Road, Zhenjiang 212013, China
\*   Correspondence: zxl1979@ujs.edu.cn

**Abstract:** In an effort to solve the issue of unadjustable damping of skyhook inertance suspension, a new adjustable device combining an inerter and a damper that aims to simultaneously adjust the inertance and damping was proposed. This article proposes a near practical mathematical model of such an adjustable device, and the model is found to be equivalent to a parallel connection of an adjustable inerter and damper. A prototype of such a device is made, and its damping and inertial forces are separated through quasi-static and dynamic mechanical character tests. The validity of the theoretical models is verified through a comparison between the test and simulation results of the mechanical character with a maximum error of 4.96% for the damping model and 6.28% for the inertial model, which lays the foundation for subsequent studies on adjustable regular patterns of inertance and damping as well as applications in semi-active ISD suspensions. In addition, the device simplifies an inerter and a damper into one device and reduces the layout space and cost, which is of great engineering application value.

**Keywords:** adjustable inertance; adjustable damping; integrated device; device modeling; rig test

## 1. Introduction

As a novel vibration absorbing element, the inerter has a great effect on the damping effect of a vibration isolation system. Therefore, in-depth study of the inerter is of great significance to reduce the vibration in the process of mechanical operation and to improve the riding comfort of the operators. In the traditional force–current analogy, the mass, damper and spring correspond to the capacitor, resistor and inductor, respectively, and the mass can only correspond to a grounded capacitor. Therefore, this loose correspondence restricts the development of the traditional force–current analogy theory.

In 2002, Smith [1] first proposed the concept of the inerter and defined it as a device at both ends. Its characteristic is that the equal and reverse force exerted at both ends is directly proportional to the relative acceleration of them. The proportional constant is called inertance, and the unit is kg. This two-terminal element is completely similar to the capacitor, which realizes the strict correspondence between the capacitor–inductor–resistor circuit network system and the inerter–spring–damper mechanical network system and allows a large number of electrical network theories and circuit analysis methods, collected over a long time, to be directly used to the analysis and comprehensive research of mechanical networks.

Inerter devices have been a main innovation in the research field of passive and semi-active vibration control [2]. In 2004, Smith [3] developed a pinion-and-rack inerter and applied it to a suspension. Compared with the traditional suspension, the ISD suspension consisting of an inerter, spring and damper improved the vehicle's vibration reducing performance. Papageorgiou et al. [4] presented the structure of a ball–screw inerter and conducted experimental tests, and the results indicated that the inerter can effectively improve the mechanical vibration reducing effect.

Wang [5] introduced an inerter to building isolation systems and effectively weakened the vibration of buildings. He [6] also investigated the inerter nonlinearities and verified that the overall performance of the vehicle suspension with an inerter having nonlinearities is still better than that of the traditional suspension, especially as the suspension has a large spring stiffness. Zhang et al. [7] first applied the semi-active shyhook damping control to the suspensions with inerters and achieved the best compromise between road holding and comfort quality.

Li et al. [8] proposed a semi-active control with adjustable inertance to track the desired control force of a fittingly designed state-feedback $H_2$ controller and showed good decrease of vibration at the sprung mass resonance frequency. Chen et al. [9] proposed a semi-active ISD suspension with LQR control, and this parallel ISD suspension equipped with an adjustable inerter and damper has better performance than the traditional semi-active suspension equipped with an adjustable damper. In 2018, they [10] proposed a novel design method including an inerter, where the whole semi-active suspension was divided into a passive part and a semi-active part. Both parts of the suspension were optimized, and the passive part was the priority. The results showed that the new design method significantly improved the overall suspension performance.

In order to respond well to the dominant period changes of ocean waves, Takino et al. [11] proposed a novel wave-energy converter (WEC) with a tuned variable inerter (TVI), which was controlled by the combination of an FFT algorithm and a motor. The proposed TVI system better responded to the wave period changes compared with a traditional tuned inerter (TI) system.

Based on magnetorheological (MR) fluid, Zhong et al. [12] proposed a new semi-active inerter and confirmed that it had the advantages of low energy consumption, rapid response and wide adjustment range. Through the above studies, it was suggested that the inerter had extensive application prospects in vibration control, particularly in the vibration control field of suspensions.

In the engineering applications of inerters, the first several inerters were mechanical, which have problems, such as friction, backlash and eccentricity [13,14], and few have been applied to racing cars [15]. To solve the above problems, Wang [16] developed an inerter with a hydraulic motor and conducted nonlinear analysis in 2009. However, the volume of such elements was large and suitable for building and bridge vibration isolation.

In 2011, Robin [17] proposed an inerter with a hydraulic piston; however, the volume was still large despite its simple structure and strong durability. In 2013, a new hydraulic inerter with a helical channel, whose volume was significantly reduced, was proposed by Swift [18], and mathematical expressions of a constant inertance and constant damping coefficient were provided.

It is worthwhile to note from the above studies that the inerter was gradually developed from mechanical elements to hydraulic elements and has a simple structure, lower cost and is free from problems, such as back clearance and eccentricity. However, restricted by their structural characteristics, it was often difficult to adjust the MOI of the above inerters; therefore, the inertance was generally constant and insufficient to satisfy the requirements of semi-active ISD suspension for inertance adjustment. In 2018, Zhang [19,20] proposed the concept of a semi-active suspension controlled by skyhook inertance based on an adjustable inerter, which had load adaptability.

Nevertheless, the lack of adjustment of damping made it impossible to adapt to road conditions. In 2020, Zhao [21] designed a displacement-dependent vibration control system called the damping inerter system for reducing seismic response, which encouraged the thought of combining an inerter and a damper. Brzeski et al. [22] proposed a new tuned mass damper with inerter (TMDI), which added a dash-pot with displacement-dependent and velocity-dependent damping coefficients. Furthermore, the maximum amplitude of the damped body can further decreased by using the non-linear dash-pot and the inerter.

Therefore, to make the suspension adapt to both load changes and road conditions, an adjustable device with a built-in helical channel that can adjust the inertance and damping

coefficient at the same time was proposed; however, the damping modeling of previous studies does not satisfy the actual situation. Based on the Hagen–Poissier flow equation, the damping coefficient of the device is an equation linearly related to the displacement $x$ [19]. In practice, the movement of the fluids in the device is affected by nonlinear factors.

The motivation of this paper is to re-establish the damping model and verify it by experiments, and a near practical equation is established after considering the movement of the fluids in the device affected by secondary flow phenomena and the pressure losses. On this basis, a prototype of the adjustable device combining an inerter and a damper is designed, and a mechanical characteristic test is performed to verify the model.

## 2. Structural Design and Working Principle

### 2.1. Basic Structure

The adjustable device combining an inerter and a damper consists of a cylinder and a valve as shown in Figure 1. In the bottom half of Figure 1, the hydraulic valve comprises a removable valve element surrounded by a helical channel on its outer surface, a control rod and a valve cylinder full of fluid. In addition, the inner surface of the right half of the valve cylinder has an larger diameter than that of the left half, which enables the working length of the helical channel to change with the movement of the valve element. In the top half of Figure 1, the hydraulic cylinder includes a movable piston, a piston rod and a cylinder full of fluid.

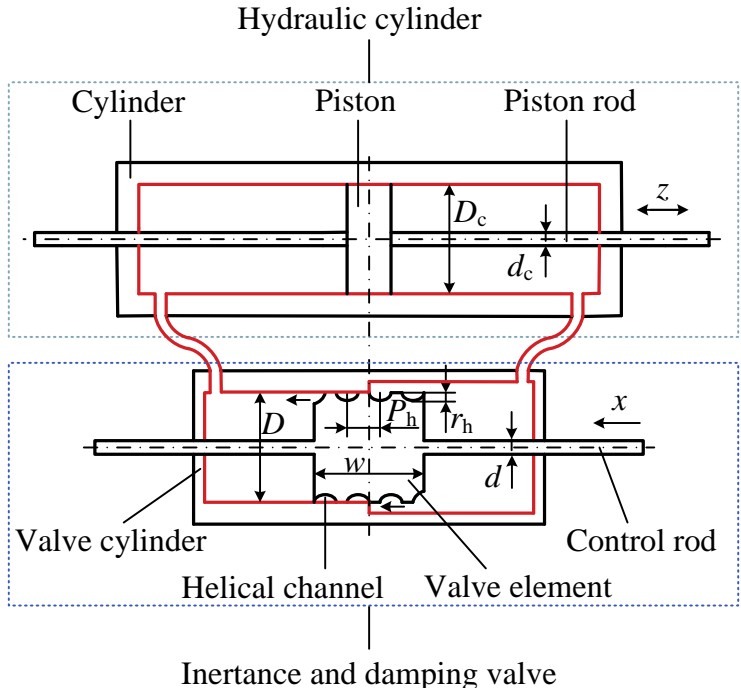

**Figure 1.** Schematic diagram of adjustable device combining an inerter and a damper.

### 2.2. Working Principle

The piston in the hydraulic cylinder drives the fluid to flow through the helical channel of the hydraulic valve; as a result, the inertial force was generated due to the variable mass of the fluid in the helical channel. At the same moment, the damping force, which mainly comes from the pressure loss along the way caused by the viscous effect of the helical channel, is also generated when the fluids flow. During the movement of the valve element, the fluid mass in the helical channel changes with the length of the helical channel, and thus the inertance and damping coefficient also change. The relative displacement between the valve cylinder and valve element driven by the control rod can be controlled continuously; therefore, the inertance and damping of the adjustable device that is proposed here can be successively varied as shown in Figure 2.

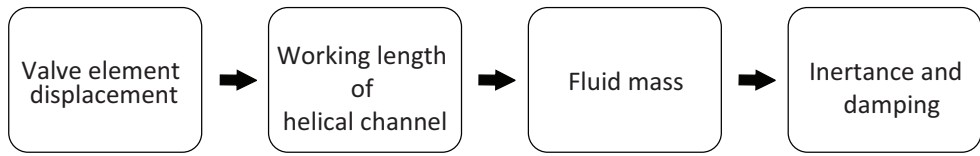

**Figure 2.** Working principles of the adjustable device combining an inerter and a damper.

### 3. Mathematical Model

#### 3.1. Assumption

To analyze the key performance, the following simplifying assumptions were made when establishing the mathematical model:

- The device is well sealed, and there is no leakage when the fluid flows.
- The fluid is incompressible and has constant density.
- The temperature, potential energy and heat loss will be ignored, and the fluid dynamic viscosity should remain unchanged.

#### 3.2. Inertance Model

According to the related calculation methods of the constant inertance of fluid [18–20,23], the ideal inertia force $f_b$ can be described by the following equation:

$$f_b = B\ddot{x} = Ba, \tag{1}$$

Assume that $B(x)$ is the adjustable inertance in kg. Based on the assumption mentioned in the previous section, $B(x)$ can be represented in the form

$$B(x) = \rho l(x) \frac{S_1^2}{A_2}, \tag{2}$$

where $\rho$ is the fluid density, $l(x)$ is the working length of the helical channel, $S_1$ is the working cross-sectional area of the piston and $A_2$ is the cross-sectional area of the helical channel.

Let $P_h$ br the helical channel pitch, $D$ be the valve element diameter, $x$ be the relative displacement between the valve cylinder and the valve element, $D_c$ be the piston diameter, $d_c$ be the piston rod diameter, and $r_h$ be the helical channel radius. Then, $l(x)$, $S_1$ and $A_2$ in Equation (2) can be expressed as

$$l(x) = \frac{\sqrt{P_h^2 + (\pi D)^2}}{P_h} x, \tag{3}$$

$$S_1 = \frac{\pi(D_c^2 - d_c^2)}{4}, \tag{4}$$

and

$$A_2 = \frac{\pi r_h^2}{2}. \tag{5}$$

Substituting Equations (3)–(5) into (2), Equation (2) can be recast as

$$B(x) = \frac{\pi\rho(D_c^2 - d_c^2)^2 \sqrt{P_h^2 + (\pi D)^2}}{8 P_h r_h^2} x. \tag{6}$$

Equation (6) suggests that the inertance at a given moment is a function of the relative displacement $x$ between two terminals and has a constant value for a set of selected parameters $D_c$, $d_c$, $D$, $P_h$, $r_h$ and $\rho$.

### 3.3. Damping Model

The pressure loss of the device proposed here mainly comes from the following three aspects:

- The pressure loss along the way generated by the flow of the viscous fluid in the helical path.
- The shear pressure loss generated by the high-pressure fluid film between the cylinder wall and the piston when the piston moves.
- The pressure loss when fluid flows through the outlet and inlet of the helical channel.

Therefore, the adjustable damping force $F_d$ is represented as

$$F_d = F_{hc} + F_s + F_{in} + F_{out}, \tag{7}$$

where $F_{hc}$, $F_s$, $F_{in}$ and $F_{out}$ are the damping forces caused by the pressure loss along the way, the shear pressure loss and the inlet and outlet pressure losses, respectively. According to the relevant calculation methods of the constant damping force of fluid [18,24,25], combining the devices in this paper, the above formulas can be obtained as

$$\begin{cases} F_{hc} = 0.03426 \dfrac{2\rho l(x) A_1}{\sqrt{D_h R_h}} \left(\dfrac{A_1}{A_2}\right)^2 \dot{z}^2 + 17.54 \dfrac{2\mu l(x) A_1^2}{D_h^2 A_2} \dot{z} \\[2mm] F_s = \dfrac{\pi \mu D x}{\Delta r} \dot{z} \\[2mm] F_{in} = 0.25 \dfrac{\rho A_1^3}{A_2^2} \dot{z}^2 \\[2mm] F_{out} = 0.5 \dfrac{\rho A_1^3}{A_2^2} \dot{z}^2 \end{cases} , \tag{8}$$

where $A_1$ is the valve element cross-sectional area, $D_h$ is the hydraulic diameter of the helical channel, $R_h$ is the bend radius, $\mu$ is the dynamic viscosity, $\dot{z}$ is the relative velocity between the two ends of the hydraulic cylinder and $\Delta r$ is the clearance between the valve element and cylinder wall.

It can be seen from reference [18] that the pressure loss along the way dominates. Therefore, the adjustable damping force can be simplified as

$$F_d = 0.03426 \frac{2\rho l(x) A_1}{\sqrt{D_h R_h}} \left(\frac{A_1}{A_2}\right)^2 \dot{z}^2 + 17.54 \frac{2\mu l(x) A_1^2}{D_h^2 A_2} \dot{z} \tag{9}$$

The effective cross-sectional area $A_1$ of the vlave element can be expressed as

$$A_1 = \frac{\pi(D^2 - d^2)}{4}, \tag{10}$$

where $d$ is the diameter of the control rod.

The hydraulic diameter $D_h$ of the helical channel is defined as

$$D_h = \frac{2\pi r_h}{\pi + 2}. \tag{11}$$

In light of Equations (3), (5) and (9)–(11), the adjustable damping coefficient is derived as

$$C(x, \dot{z}) = c_1(x)\dot{z} + c_2(x), \tag{12}$$

where

$$c_1(x) = 0.0042825 \frac{\pi \rho (D_c^2 - d_c^2)^3 \sqrt{P_h^2 + (\pi D)^2}}{P_h r_h^4 \sqrt{D_h R_h}} x,$$

$$c_2(x) = 4.385 \frac{\pi \mu (D_c^2 - d_c^2)^2 \sqrt{P_h^2 + (\pi D)^2}}{P_h r_h^2 D_h^2} x.$$

It is observed from Equations (6) and (12) that the damping coefficient and the inertance are both functions of the valve element displacement. Therefore, in the designed device, the damping and inertance can be simultaneously adjusted by changing the displacement of the valve element, which means that this device is equivalent to an inerter in parallel with a damper as shown in Figure 3.

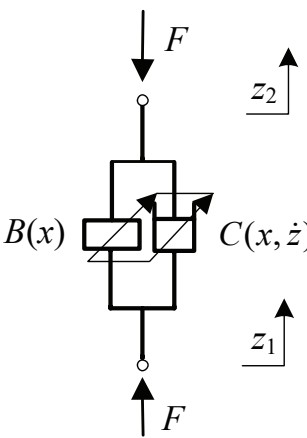

**Figure 3.** Equivalent model of the adjustable device combining an inerter and a damper.

## 4. Test and Results

### 4.1. Prototype Device

To verify the inertance model and damping model represented by Equations (6) and (12), considering factors, such as cost and processing technology, the prototype of the designed adjustable device and the valve element are made as shown in Figure 4. The device parameter details are shown in Table 1.

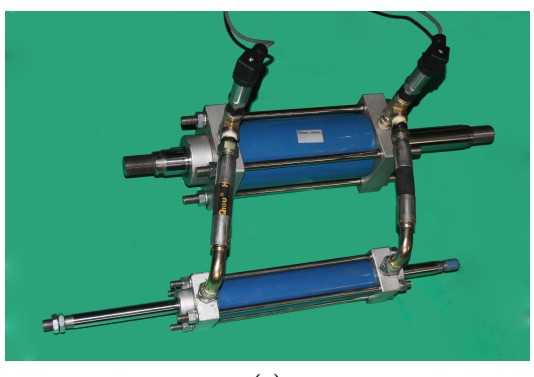
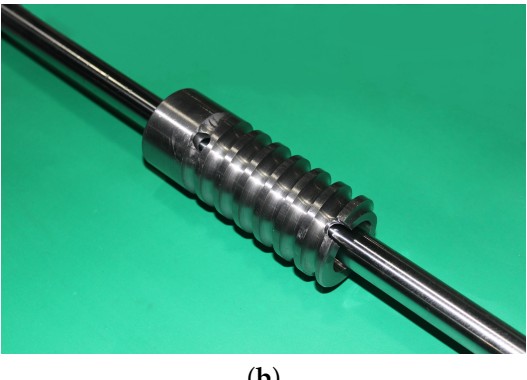

| (a) | (b) |

**Figure 4.** Prototype of the device combining an adjustable inerter and a damper. (**a**) Prototype device. (**b**) Valve element with a helical channel.

**Table 1.** The parameters of the adjustable device combining an inerter and a damper.

| Description | Value |
| --- | --- |
| Piston diameter $D_c$ | 0.1 m |
| Piston rod diameter $d_c$ | 0.04 m |
| Valve element diameter $D$ | 0.05 m |
| Control rod diameter $d$ | 0.01 m |
| Helical channel radius $r_h$ | $5.4 \times 10^{-3}$ m |
| Helix pitch $P_h$ | 0.014 m |
| Valve element width $w$ | 0.1 m |
| Fluid dynamic viscosity $\mu$ | $4.94 \times 10^{-4}$ Pa s |
| Fluid density $\rho$ | 760 kg m$^{-3}$ |

### 4.2. Experimental Instruments and Layout

A test rig was used for testing the device prototype as demonstrated in Figure 5. The actuator displacement in the test rig is controlled by an Instron 8800 vibration excitation system, and the prototype to be tested is placed between the loading device (fixed to beam) and the hydraulic actuator. The load cell and LVDT sensors inside the actuator provide measurements of the load force through the prototype and the actuator displacement, respectively. In addition, two MIK-P300 pressure sensors are used for precisely measuring the pressure of the fluid inside the lower and upper chambers. All the measurement data were recorded with the SCM05 acquisition system.

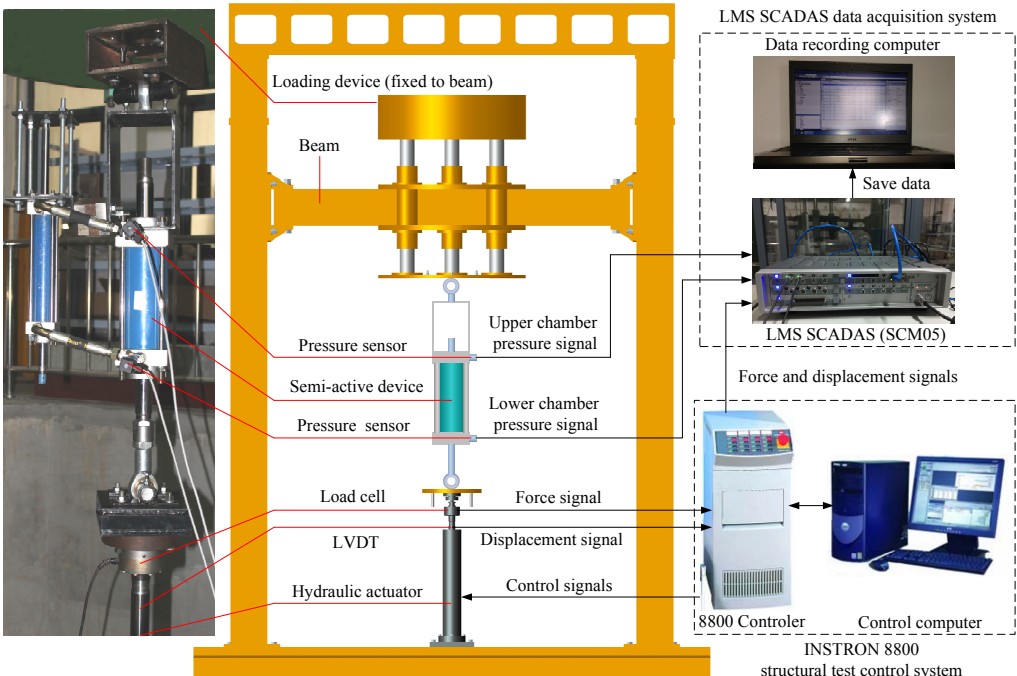

**Figure 5.** Test rig for prototype device combining an adjustable inerter and a damper.

### 4.3. Friction Separation

According to relevant experimental studies [18,26], the damping and inertial force of the adjustable device can accurately separated from the total force by displacement excitation of different waveforms. The output force of the adjustable device combining an inerter and damper mainly comprises inertial force $F_i$, damping force $F_d$ and friction force $F_f$ between the inner wall of the cylinder and the piston of the hydraulic cylinder as shown in Figure 6 and Equation (13).

$$F = F_i + F_d + F_f, \tag{13}$$

among which,

$$F_i = B(x)\ddot{z}, \tag{14}$$

$$F_d = c_1(x)\dot{z}^2 + c_2(x)\dot{z}, \tag{15}$$

$$F_f = f\,\mathrm{sign}(\dot{z}), \tag{16}$$

where $f$ refers to the maximum static friction.

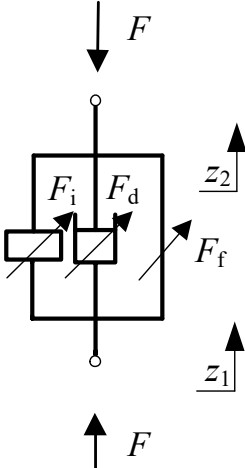

**Figure 6.** The output force of the adjustable device combining an inerter and a damper.

First, the quasi-static characteristic test is used to separate the friction. A triangular wave with frequency $f = 0.01$ Hz is selected as the displacement input in the test scheme so that the piston is in a quasi-static state (the speed is less than 0.01 m/s) during the whole test process during the whole test. At this time, the acceleration and velocity were approximately zero, and thus the inertial force and the damping force could be ignored. Therefore, when the motion is stable in each cycle, the output resultant force measured by the force sensor of the excitation table can be approximately the friction. The experimental value of the force of friction and the fitting curve are shown in Figure 7. As shown in Figure 7, the friction force of the adjustable device combining an inerter and a damper can be estimated as 240 N.

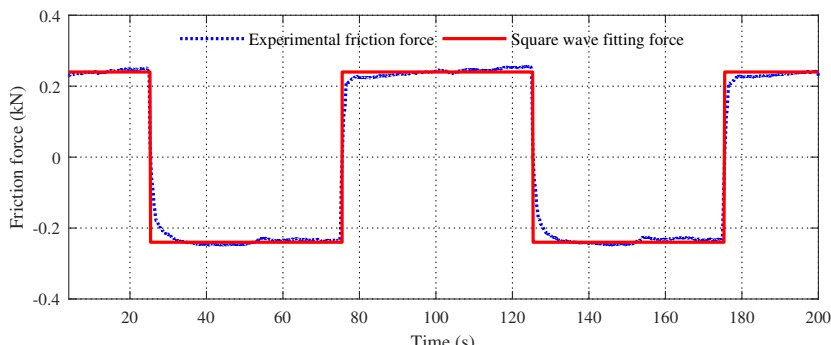

**Figure 7.** The force of friction under the quasi-static test.

### 4.4. Damping Force Separation

To separate the damping force, a dynamic characteristic test with a triangular wave as the displacement input was conducted. Triangular waves with frequencies of 0.17, 0.42, 0.83, 1.67 and 2.50 Hz were selected as the test scheme. Under uniform motion, the acceleration was approximately zero; therefore, the inertial force can be ignored. The pressure values of the two chambers of the hydraulic cylinder were measured by the pressure sensor, and the pressure difference between the lower and upper chambers was calculated as $\Delta p$. On this basis, the damping force can be calculated, which is

$$F_{\mathrm{d}} = \triangle p S_1$$

Under excitation amplitudes of 30 and 40 mm, a long-term steady state is generated, which makes the force values measured at both ends more accurate. Note that since the output force at 1.67 Hz exceeds the maximum bearing capacity of the shaking table,

triangular waves with an amplitude of 20 mm and a frequency of 1.67 Hz and waves with an amplitude of 10 mm and a frequency of 2.5 Hz were selected to represent high-frequency performance.

Moreover, the valve element of the control valve is adjusted to the chamber with an enlarged diameter of the inner surface before the test. At this time, the helical channel does not work, which means that the damping and inertance coefficients are 0. In the test, according to the different experimental conditions, when the valve element of the regulating valve is moved once every 10 mm relative to the test starting position, the working length of the helical channel changes, and the values of inertance and damping coefficient also change accordingly. The adjustable range of the valve element displacement is 0~100 mm.

To reduce the influence of test error caused by accidental factors, tests in each working condition were conducted twice, and the results were compared to improve the accuracy. Since there are 80 kinds of test conditions, due to the limited space, the fitting curves of test values under certain working conditions were generally selected for presentation. Taking the valve element displacement $x$ = 50 mm as an example, the identified damping force is shown in Figure 8.

Furthermore, the experimental and theoretical values of the damping force and velocity corresponding to six working conditions under this valve element displacement are shown in Figure 9. The test values are essentially consistent with the theoretical values, which means the new mathematical model proposed in Section 3.3 can provide an accurate representation of the actual damping of the adjustable device.

Furthermore, the relationship among the damping coefficient, the valve element displacement and the relative velocity at both ends of the hydraulic cylinder is shown in Figure 10. As shown in the figure, the test data essentially agree with the theoretical data, and the maximum relative error is 4.96%, which demonstrates the consistency between the theoretical model and the test data as well as the high accuracy of the damping model of the adjustable device. It is worthwhile to note that the error between the experimental data and theoretical data is large at 0.17 Hz. The reason is that the signal-to-noise ratio in the low-frequency band is small, which increases the measurement error; however, the maximum error is less than 5%, which can meet the project requirements.

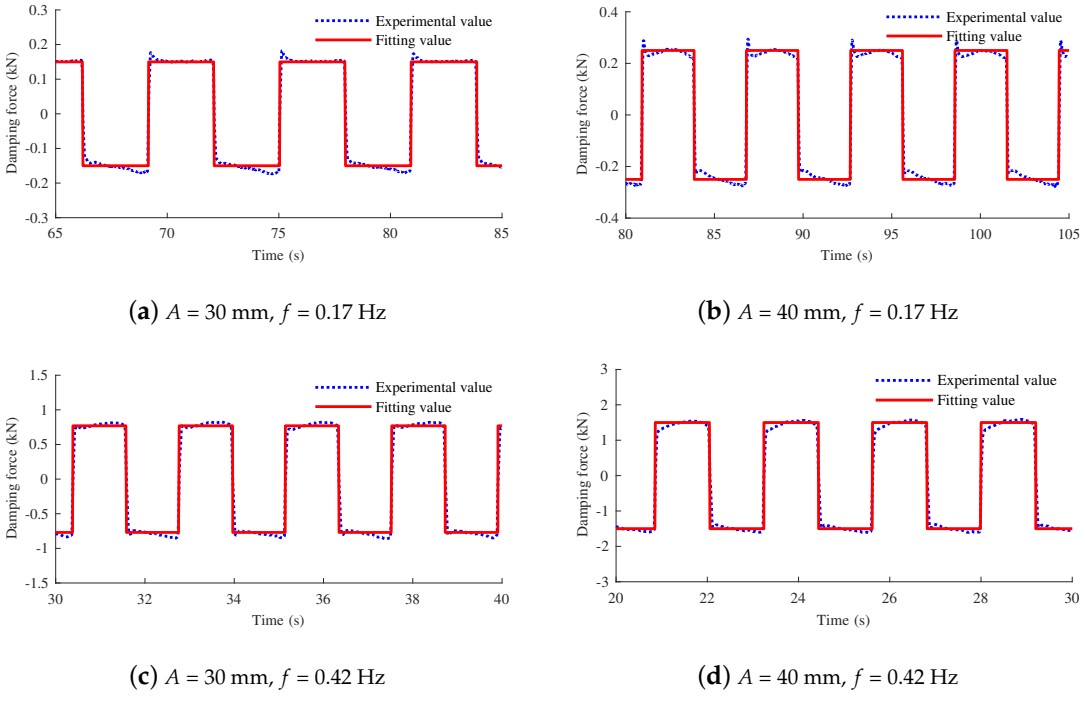

(**a**) $A$ = 30 mm, $f$ = 0.17 Hz                  (**b**) $A$ = 40 mm, $f$ = 0.17 Hz

(**c**) $A$ = 30 mm, $f$ = 0.42 Hz                  (**d**) $A$ = 40 mm, $f$ = 0.42 Hz

**Figure 8.** *Cont.*

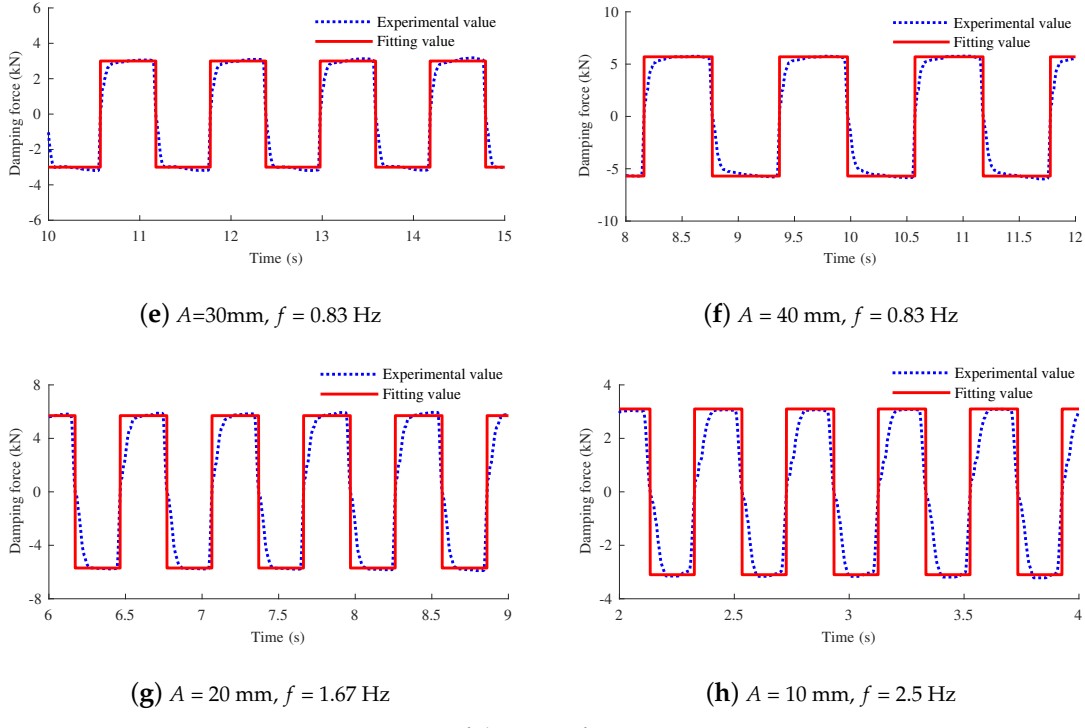

(**e**) *A* = 30mm, *f* = 0.83 Hz

(**f**) *A* = 40 mm, *f* = 0.83 Hz

(**g**) *A* = 20 mm, *f* = 1.67 Hz

(**h**) *A* = 10 mm, *f* = 2.5 Hz

**Figure 8.** Fitting curves of damping force.

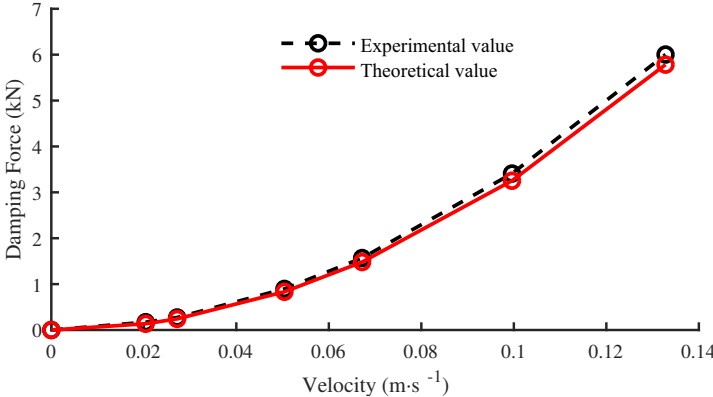

**Figure 9.** Damping characteristic curve.

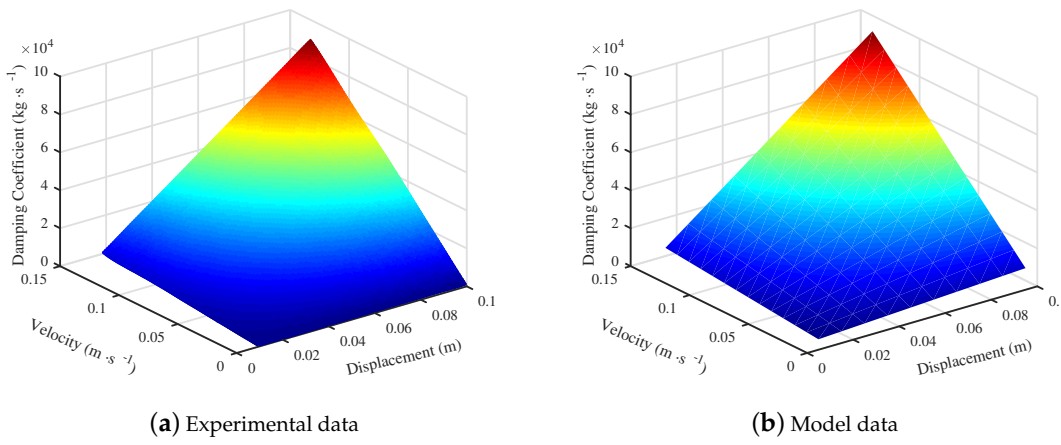

(**a**) Experimental data

(**b**) Model data

**Figure 10.** Damping characteristic surface.

### 4.5. Inertial Force Separation

The sinusoidal displacement input is used in the dynamic performance test of the device to separate the inertial force. Limited by the maximum bearing capacity, the test scheme chose sine waves with an amplitude of 30 mm and frequencies of 0.17, 0.42 and 0.83 Hz, an amplitude of 20 mm and a frequency of 1.67 Hz and an amplitude of 10 mm and a frequency of 2.5 Hz as the input. The steps of separating the inertial force are as follows:

- Based on the displacement signal, valve element displacement and damping force identified in Section 4.4, the test values of the proportional coefficients $C_1$ and $C_2$ in the damping coefficient are identified according to Equation (15).
- The displacement signal measured in this section is substituted into Equation (15) to obtain the test value of the damping force.
- The inertial force test value can be obtained by subtracting the friction force and damping force test value from the measured output resultant force.

There are a total of 50 test data processing conditions; however, due to limited space, this paper extracted some representative test data processing results. Take the valve element displacement $x$ = 30 mm as an example. The experimental and theoretical values of the output resultant force, damping force and inertial force of the device are shown in Figures 11–13. The curve trends of the test value and the theoretical value of the device's output force are roughly the same at 0.17 Hz, and are manifested as the coupling of square waves and triangular waves.

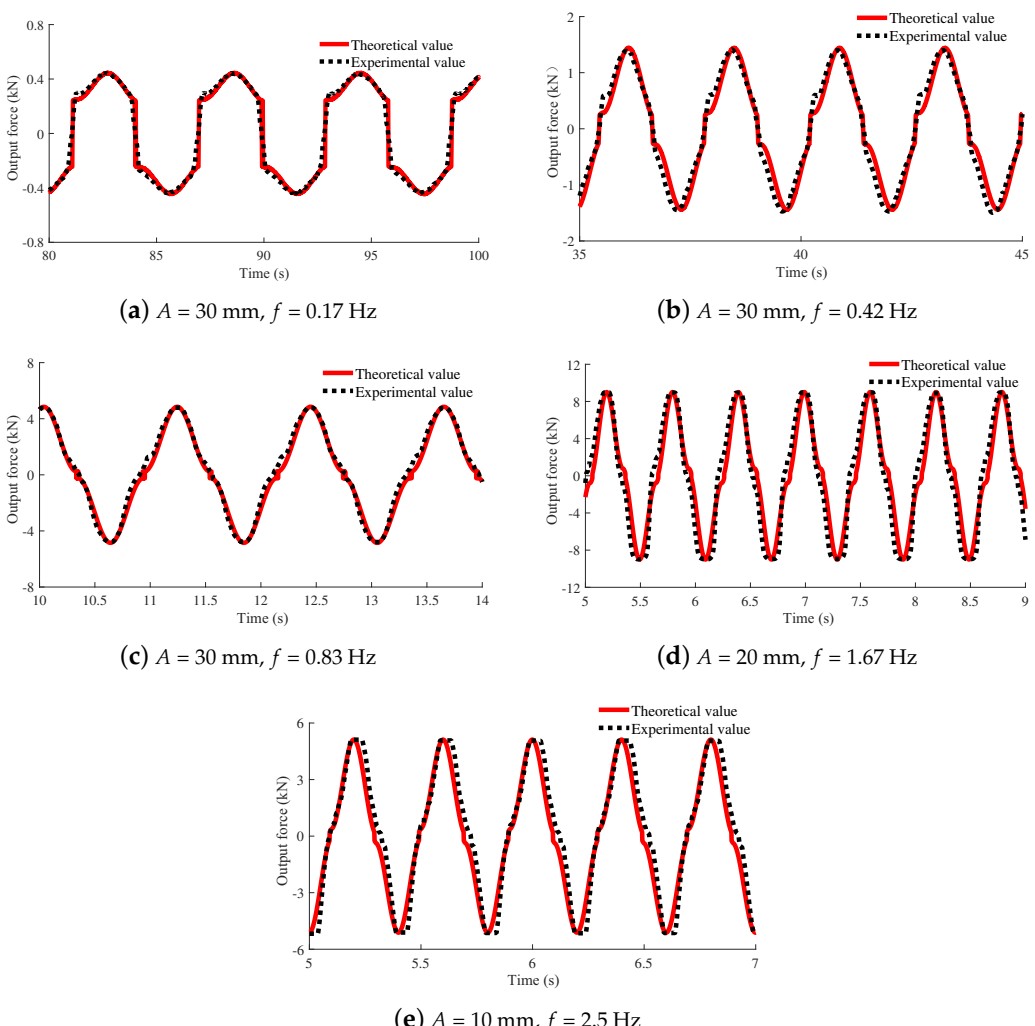

(**a**) $A$ = 30 mm, $f$ = 0.17 Hz

(**b**) $A$ = 30 mm, $f$ = 0.42 Hz

(**c**) $A$ = 30 mm, $f$ = 0.83 Hz

(**d**) $A$ = 20 mm, $f$ = 1.67 Hz

(**e**) $A$ = 10 mm, $f$ = 2.5 Hz

**Figure 11.** Comparison of the experimental and theoretical values of the total output force for an adjustable device.

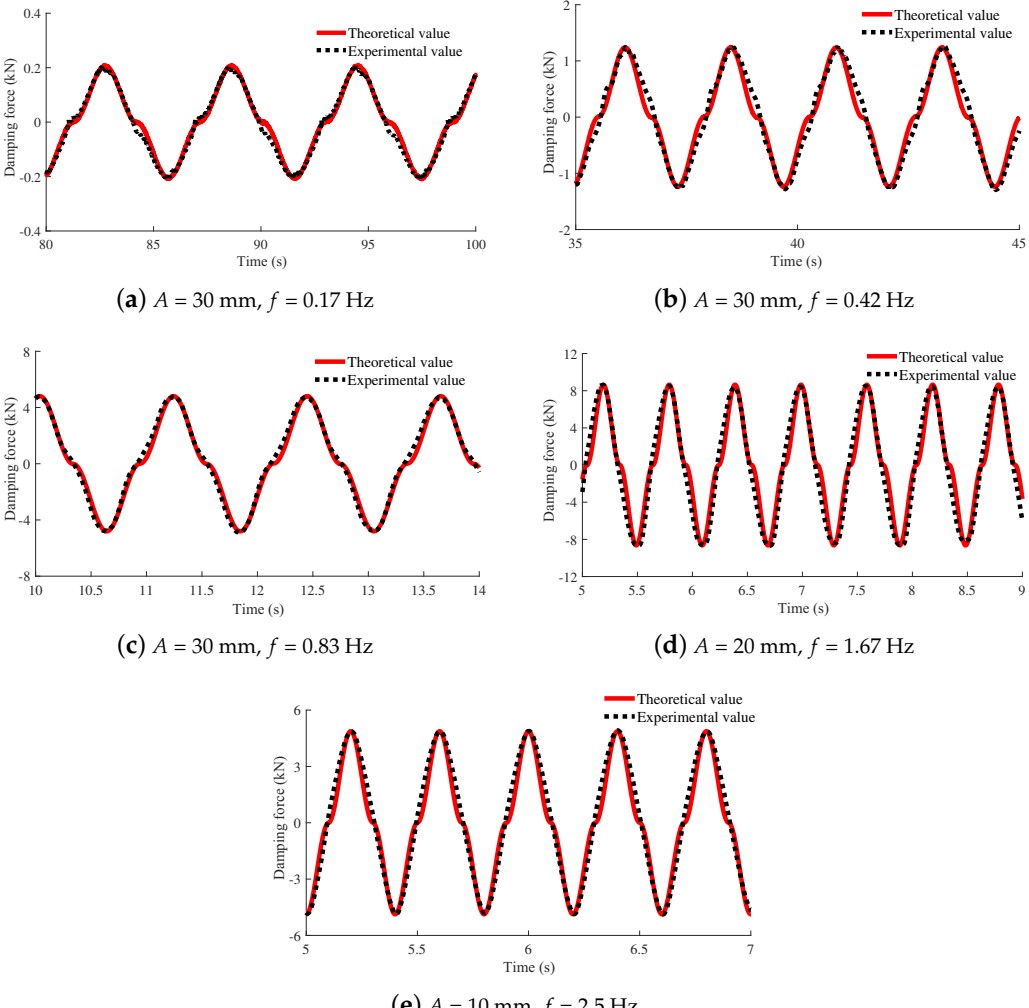

(**a**) $A$ = 30 mm, $f$ = 0.17 Hz

(**b**) $A$ = 30 mm, $f$ = 0.42 Hz

(**c**) $A$ = 30 mm, $f$ = 0.83 Hz

(**d**) $A$ = 20 mm, $f$ = 1.67 Hz

(**e**) $A$ = 10 mm, $f$ = 2.5 Hz

**Figure 12.** Comparison of the experimental and theoretical values of the damping force for an adjustable device.

At low frequencies, friction is dominant; with increasing frequency, the damping force and inertial force gradually increase and play a major role. At this time, the experimental values of the output resultant force, damping force and inertial force of the device essentially match the theoretical values, all of which are generally sinusoidal. The results demonstrated that the method of separating the above inertial forces is reliable.

According to the above test data, the relationship between the inertance and valve element displacement under all valve element displacements can be obtained as shown in Figure 14 and Table 2. It is worth noting that the experimental and theoretical values of inertance are essentially consistent, and the maximum relative error is 6.28%, which confirms the correctness of the inerter model.

At the same time, the correctness of the damping model is verified. In addition, the error between the experimental and theoretical values of the inertance is large when the valve element displacement is large, which may result from the decrease in the fluid density caused by the increase in the working fluid temperature when the output force is large. However, the relative error is still less than 7%, which is acceptable in engineering applications.

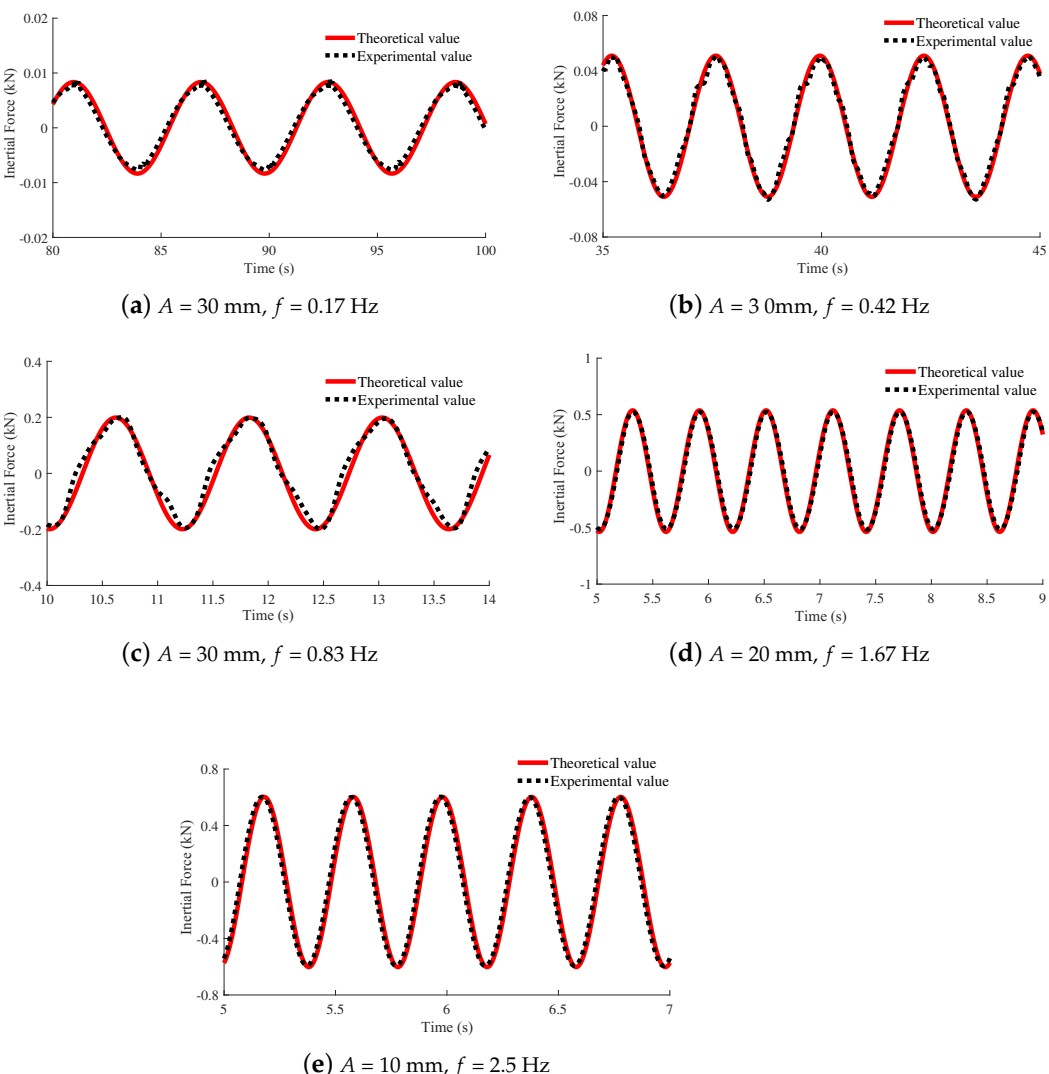

(**a**) $A$ = 30 mm, $f$ = 0.17 Hz

(**b**) $A$ = 3 0mm, $f$ = 0.42 Hz

(**c**) $A$ = 30 mm, $f$ = 0.83 Hz

(**d**) $A$ = 20 mm, $f$ = 1.67 Hz

(**e**) $A$ = 10 mm, $f$ = 2.5 Hz

**Figure 13.** Comparison of the experimental and theoretical values of the inertial force for an adjustable device.

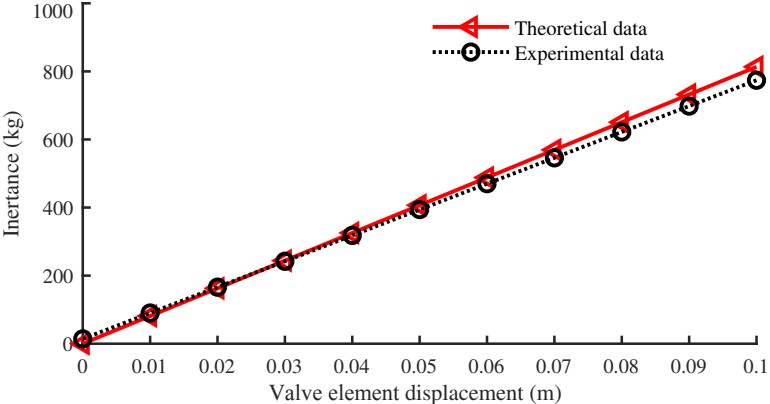

**Figure 14.** Comparison of curves between the experimental and theoretical inertances.

**Table 2.** Inertance under different valve element displacements.

| Valve Element Displacement (mm) | Experimental Value (kg) | Theoretical Value (kg) | Error (%) |
| --- | --- | --- | --- |
| 10 | 84.0 | 81.3 | 3.28 |
| 20 | 168.2 | 162.7 | 3.40 |
| 30 | 249.8 | 244.0 | 2.34 |
| 40 | 321.3 | 325.4 | −1.27 |
| 50 | 407.4 | 406.7 | 0.17 |
| 60 | 478.8 | 488.1 | −1.90 |
| 70 | 549.4 | 569.4 | −3.52 |
| 80 | 622.5 | 650.8 | −4.25 |
| 90 | 691.1 | 732.1 | −5.61 |
| 100 | 764.4 | 813.5 | −6.28 |

## 5. Conclusions

In this study, a near practical mathematical model of the adjustable device was built. Considering the effects of nonlinear factors, the damping coefficient of the device was modeled as a nonlinear equation related to the relative displacement as well as the relative velocity. On this basis, a trial prototype device was made. The friction force, damping force and inertial force were separated through quasi-static and dynamic mechanical characteristic tests, and the damping model and inertance model were fitted. The test results and simulation were compared and analyzed. The results showed that:

- At low frequency, friction plays the main role. However, with increasing frequency, the proportion of inertial force and damping force increases and occupies the dominant position; at this moment, the device is equivalent to the parallel structure of an inerter and a damper.
- The maximum error between the experimental and theoretical values of the damping model was 4.96%, and that of the inertance model was 6.28%, which indicates that the theoretical model essentially matches the actual situation; hence, the designed device is feasible and meets the needs of engineering applications.
- The inertance of this device relates to the displacement of the valve element. Moreover, its damping relates to both the valve element displacement and the relative velocity at the terminals of the hydraulic cylinder. As a result, the damping and inertance can be simultaneously adjusted by controlling the displacement of the valve element.

In conclusion, the adjustable device combining an inerter and a damper satisfied the requirements for adjusting inertance and damping. Moreover, the inerter and damper were simplified into one device, reducing the layout space and the cost, which has good engineering application value. As a next step, considering that the error cannot be ignored when the frequency is low and the output force is large, we will attempt to find a better test method to reduce the test error and improve the model accuracy.

Then, based on the adjustable model combining an inerter and a damper, we will apply this to the field of semi-active suspensions. Due to the simplified design, the device is ideally suited for skyhook semi-active suspension applications, where the suspension damping and inertial forces can be changed simultaneously by a single control system. In future research, the dynamic model of an ISD suspension system will be established, and further simulation and experiments will be performed.

**Author Contributions:** Conceptualization, X.Z. and W.Z.; methodology, X.Z.; software, W.Z.; validation, X.Z., W.Z. and J.N.; formal analysis, X.Z.; investigation, J.N.; resources, X.Z.; data curation, X.Z.; writing—original draft preparation, X.Z.; writing—review and editing, X.Z.; project administration, X.Z.; funding acquisition, X.Z. All authors have read and agreed to the published version of the manuscript.

**Funding:** This research was funded by the National Natural Science Foundation of China,grant number 51875257.

**Institutional Review Board Statement:** Not applicable.

**Informed Consent Statement:** Not applicable.

**Data Availability Statement:** Data sharing not applicable.

**Acknowledgments:** The authors are grateful to the editor and anonymous reviewers for their constructive comments and suggustions, which have improved this paper.

**Conflicts of Interest:** The authors declare no conflict of interest.

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
