# Peer review of "Modelling and Experiment of an Adjustable Device Combining an Inerter and a Damper"

_machines, doi:10.3390/machines10090807_

Round 1

Reviewer 1 Report

This manuscript deals with the numerical simulation and experimental test of the compact inerter-damper with displacement-dependent inertance and damping coefficients. Given the valuable contribution of the experiments, the topic is of interest for developing new devices for vibration control. Some detailed comments should be followed for revision before publication:

(1)   As this compact device has been proposed by the author in a previous study, the novelty and major contribution should be stressed in the Abstract, Introduction, and Conclusion. The relationship between the current manuscript and the previous one should be clearly stated by focusing on the experiment results.

(2)   Please justify the definition of ‘adjustable device’ as the inertance and damping coefficient cannot be changed if the device is manufactured, which differs from the semi-active dampers using the adjustment approach. The usage of ‘displacement-dependent’ could be suitable.

(3)   In the introduction, the recent advances in the inerter-damper with variable damping or inerters should be added and commented.

https://doi.org/10.3390/en13236175

https://doi.org/10.1002/stc.2890

https://doi.org/10.1016/j.oceaneng.2022.111696

(4)   For easier reference, a more detailed explanation of the mechanical model and numerical simulation of the inertance and damping should be added in Section 3.

(5)   The data processing and filtering method should be explained for the experiment results in Section 4. Referring to the experiment data recording methods mentioned in previous studies, using damping support in series with the inerter could reduce the recording error. Please explain the reason for the current experiment setting by commenting on this issue.

Author Response

Thanks to the editor and experts for your valuable comments and suggestions on our work,for the comments and suggestions put forward by editor and experts, we have modified our manuscript accordingly.
The attachment includes the response to editor and experts. Please see the attachment.

Reviewer 2 Report

The paper is interesting and I do not have comments on the paper's modelling and experimental part. However, there is a lack of references to papers of Brzeski/Perlikowski and MZQ Chen on inerter with variable inertance.  Please expand the introduction.

Author Response

Point 1: There is a lack of references to papers of Brzeski/Perlikowski and MZQ Chen on inerter with variable inertance.  Please expand the introduction.

Response 1: We thank the reviewer for this recommend comment. We have cited three recent advances in the field of variable inerter in revised manuscript. In the second paragraph of the introduction, we cited ‘Inerter-based semi-active suspensions with low-order mechanical admittance via network synthesis’[https://doi.org/10.1177/0142331217744852] and commented it in the manuscript as ’In 2018, they proposed a novel design method include inerter, where the whole semi-active suspension is divided into a passive part and a semi-active part. Both two part of suspension was optimized respectively and passive part is in priority. The results showed that new design method can significantly improve the overall suspension performance.’ 
In the same paragraph we also cited ‘Numerical study of a point absorber wave energy converter with tuned variable inerter’[https://doi.org/10.1016/j.oceaneng.2022.111696] and commented it in the manuscript as ’In order to respond well to the dominant period changes of ocean waves, Takino et al. proposed a novel wave energy converter(WEC) with a tuned variable inerter(TVI), which is controlled by the combination of an FFT algorithm and a motor. The proposed TVI system had better responded to the wave period changes than traditional tuned inerter(TI) system.’ 
Besides, we cited ’Influence of variable damping coefficient on efficiency of TMD with inerter’[https://doi.org/10.3390/en13236175] in the fourth paragraph in introduction and commented it as ‘Brzeski et al. proposed a new tuned mass damper with inerter(TMDI), which added a dash-pot with displacement-dependent and velocity-dependent damping coefficient. And the maximum amplitude of the damped body can further decreased by using the non-linear dash-pot and the inerter.’

Round 2

Reviewer 1 Report

The manuscript has been revised according to the reviewer's comments.